# The Secret Garden of Neuronal circRNAs

**DOI:** 10.3390/cells9081815

**Published:** 2020-07-31

**Authors:** Silvia Gasparini, Valerio Licursi, Carlo Presutti, Cecilia Mannironi

**Affiliations:** 1Department of Biology and Biotechnology Charles Darwin, Sapienza University of Rome, 00185 Rome, Italy; silvia.gasparini@uniroma1.it (S.G.); valerio.licursi@uniroma1.it (V.L.); 2Institute of Molecular Biology and Pathology, National Research Council, 00185 Rome, Italy

**Keywords:** circRNA, ncRNA, miRNA, gene regulation, nervous system

## Abstract

High-throughput transcriptomic profiling approaches have revealed that circular RNAs (circRNAs) are important transcriptional gene products, identified across a broad range of organisms throughout the eukaryotic tree of life. In the nervous system, they are particularly abundant, developmentally regulated, region-specific, and enriched in genes for neuronal proteins and synaptic factors. These features suggested that circRNAs are key components of an important layer of neuronal gene expression regulation, with known and anticipated functions. Here, we review major recognized aspects of circRNA biogenesis, metabolism and biological activities, examining potential new functions in the context of the nervous system.

## 1. Introduction

circRNAs are a class of single-stranded RNA molecules that are covalently closed and lack a 5′ cap and 3′ poly(A). They were discovered by Sanger et al. in a virus infecting higher plants [1] and shortly afterward by Coca-Prados [2], who were able to detect them by electron microscopy in the eukaryotic cell cytoplasm. In 1991, Nigro et al. reported the formation of a circular RNA from an incorrect splice event of the pre-mRNA of the tumor suppressor gene Deleted in Colorectal Cancer (DDC) [3]. Although this kind of incorrect splice events was confirmed by other reports [4,5], circRNAs have long been considered to be byproducts of RNA splicing and not analyzed in detail. In the recent years, progresses in sequencing techniques and the development of dedicated bioinformatic algorithms led to the identification of a large number of circRNAs in human cells [6]. Since then, the number of identified circRNAs in eukaryotic cells has grown continuously. In 2013, Jeck et al. demonstrated the existence of up to 2500 of them in human fibroblast [7], Memczak et al. described about 2000 circRNAs in human cell lines, many of them conserved in mice and nematodes [8]. Werfel et al. estimated at least 9000 circRNAs in many different organisms [9]. Taken together, all these data clearly indicate that circRNAs are conserved and widely expressed in almost all eukaryotic organisms [10].

Remarkably, several studies in flies and mammals have demonstrated that circRNAs are preferentially enriched in neuronal tissues [8,11,12,13]; they are abundant in embryonic tissues and regulated during neuronal differentiation, nervous system development and aging [12,13,14,15] Their synaptic enrichment and their activity-dependent regulation strongly support key roles of circRNAs in the nervous system [12,13,16].

In this review, we present an overview of the current knowledge of circRNA biogenesis and functions, indicating established experimental strategies for their study. In the second part of the review, we focus on neuronal circRNAs, describing their specific features and their known dysregulation in human neuronal disorders. The identification by in silico analysis of high-energy interactions between synaptic circRNAs and microRNAs (miRNAs), prompted us to hypothesize a miRNA degradation mechanism mediated by neuronal circRNAs.

## 2. Biogenesis and Degradation of circRNAs

A large number of endogenous circRNAs have been identified in eukaryotic cells, which are stable and abundant, present both in the nucleus and in the cytoplasm. They show regulated expression both temporally, during development and differentiation, and spatially, in nuclear-cytoplasmic, soma-distal, and in different tissues [12,13].

The mechanism by which most circRNAs are generated depends on non-canonical splicing events, the so-called backsplicing, in which the 3′ end of a downstream exon is joined to the 5′ end of an upstream exon [7]. Backsplicing often competes and interferes with canonical splicing, therefore common signals and factors regulate the two processes. Based on sequence composition, circRNAs have been classified in three main classes: exonic (ecircRNAs/E-circRNAs), intronic (ciRNAs) and exonic-intronic (EIciRNAs) [6,7,17,18] (see Figure 1). In the nervous system, circRNAs are essentially exonic. Two models for exonic circRNA production have been proposed: the first is called “lariat-driven”, in which a canonical splicing event leads to the formation of a lariat intermediate, that is subsequently backspliced into a circRNA. The following elimination of intronic sequences generates the mature exonic circRNA. The second model is called “intron pairing-driven” circularization, accordingly, flanking intronic sequences in the pre-mRNA form a circle by complementation. The circle is then backspliced and the intronic sequences degraded. The two mechanisms co-exist in the cell, while the latter is probably the most common one [19]. The formation of intronic and exonic-intronic circRNAs is less well characterized. Intronic circRNAs can be considered a byproduct of the canonical linear splicing, since they derive from the escape of the intra-lariats from the enzymatic debranching activity [18,20]. Exonic-intronic circRNAs might derive from the retention of an intervening intron during the formation of an RNA circle containing more than two exons [21]. Some sequences related to the formation of such intermediates have been identified and, in general, intronic sequences able to pair to each others are thought to be necessary and important for the formation of circRNAs [18]. In this regard, longer flanking introns may be rich in complementary repeats, in turn promoting a highly efficient exon circularization. Alu repeats, sequences of about 300 nucleotides, are very abundant in introns of primate genomes, and are good examples of sequences involved in this regulation [7,22]. Indeed, the secondary structure of the pre-mRNA seems to be responsible for the fate of the pre-mRNA itself: alternative formation of secondary structures and competition between them may regulate alternative circularization and generate different circRNAs from the same gene [6,7,8,13,23]. On the other hand, the formation of secondary structures does not necessarily indicate the possibility of circularization, for instance a highly stable hairpin between repeats may inhibit circRNA biogenesis [24]. At the same time, some pre-mRNAs that do not present flanking intronic complementary repeats, may still generate circRNAs, implying that other factors are involved in their biogenesis. Many different RNA-binding proteins (RBPs) bind to pre-mRNAs to regulate alternative splicing (AS), including the formation of circular molecules [25,26]. Ashwal-Fluss et al. in 2014 demonstrated that the Muscleblind protein (MBNL) binds to specific sites in introns to stabilize cyclic structures, thus facilitating the biogenesis of circRNAs [25]. In 2015, Conn et al. showed that the induced expression of the RBP Quaking (QKI) during the epithelial–mesenchymal transition is accompanied by an increase in circRNA level, whereas the silencing of QKI leads to a decrease in circRNA abundance [26]. The authors demonstrated a role of QKI on circRNA biogenesis by its binding to flanking intronic sequence motives. Interestingly, QKI responsive elements are sufficient to induce de novo circRNA formation from transcripts linearly spliced by default mechanisms. Furthermore, RBPs are able to destabilize secondary structures in the pre-mRNA that interfere with circRNA formation. For instance, the silencing of type 1 double-stranded RNA-specific adenosine deaminase (ADAR1), results in an increase in the expression of various circRNAs in mouse and human neuronal cells [12,27].

The mechanisms underlying circRNAs turn-over and degradation are still poorly understood. Due to the lack of 5′ and 3′ free ends, circRNAs are extremely resistant to the action of RNA degradation enzymes that process terminal nucleotides. In fact, the half-life of many circRNAs has been measured to be around 48 h and up to several days in some cases [23]. Considering their high stability, how do cells regulate the concentration of circRNAs? Hansen et al. in 2011 reported that the degradation of the circRNA CDR1as/ciRS-7 (from here on, referred to as CDR1as) is achieved through a very specific pathway based on an almost perfect complementarity of the miRNA miR-671 and its target sequence present in CDR1as [28,29,30]. It is likely that other mechanisms involving endonucleolytic cleavage are still to be discovered.

Very recent studies showed that circRNAs respond to viral infections and participate in the antiviral host response [30,31]. Liu and co-authors demonstrated that, upon viral infection, circRNAs are rapidly and massively degraded by the cellular nucleic acid receptor RNase L. This process is required for Protein Kinase-R (PKR) activation in early cellular innate immune responses [32]. On the other hand, the intracellular levels of circRNAs could be modulated by exocytosis events. In fact, circRNAs have been isolated from export vesicles, like exosomes or microvesicles [32,33,34].

## 3. circRNA Known Functions

### 3.1. Regulation of Gene Transcription and Splicing

Nuclear circRNA molecules can act as regulators of their host genes expression. Through their complementarity to the parental genomic DNA, they might modulate and influence the processes of transcription and pre-mRNA splicing [18,21]. For instance, Zhang et al. identified in human cell lines several ciRNAs localized near the sites of their synthesis and strongly associated with the RNA Pol II elongation complex [18]. Further, they characterized the abundantly expressed ci-ankrd52, that once produced, interacts with the phosphorylated form of RNA Pol II and increases its host gene transcription efficiency. A decrease in mRNA synthesis is associated with ci-ankrd52 knockdown (KD) probably because of the concurrency of splicing and transcription events [18]. Similarly, the EIciRNAs, EIciEIF3J and EIciPAIP2 localize at the promoters of their parent genes and are associated with RNA Pol II [21]. Moreover, EIciRNAs were able to bind the U1 small nuclear ribonucleoprotein (snRNP), forming a complex implicated in the induction of host gene transcription.

circRNAs, by interacting with their parental locus, can decrease the activity of RNA pol II and induce transcriptional pausing. Conn et al. studied SEP3 exon6 circRNA, a nuclear circRNA derived from the exon 6 of the SEPALLATA3 gene (SEP3): in Arabidopsis, the overexpression (OE) of SEP3 exon6 circRNA promotes the expression of the cognate linear transcript lacking exon 6. The authors observed that SEP3 exon6 circRNA, by binding to the genomic parental DNA, forms an R loop (RNA:DNA hybrid), which, in turn, causes the stalling of RNA Pol II, then promoting the recruitment of splicing factors and AS [35].

Moreover, circRNAs can affect gene expression by acting as modulators of chromatin epigenetic status. The FECR1 circRNA positively regulates the expression of its host gene FL1 [36]. By immunoprecipitation assays, Chen et al. demonstrated that FECR1 is able to bind the FL1 promoter and to recruite the TET1 demethylase enzyme, eventually activating FL1 transcription. This nuclear circRNA was also found to be associated to the DNMT1 methylase promoter, inducing its downregulation. By acting as a modulator of DNA methylating and demethylating enzymes, FECR1 activates a positive feedback mechanism that boosts the transcription of FLI1 and stimulates its role in cancer invasiveness.

### 3.2. circRNAs as miRNA Sponges

circRNAs can act as functional competing endogenous RNA (ceRNA) molecules by binding to miRNAs and interfering with their functions.

To date, the best example of a circRNA acting as miRNA sponge was described by Hansen et al., who studied the regulatory interaction between CDR1as and miR-7 [37]. Human CDR1as sequence contains >70 evolutionarily conserved miR-7 binding sites, the circRNA/miRNA interaction was confirmed by AGO PAR-CLIP [8] and HITS-CLIP experiments [37]. CDR1as KD, in colorectal cancer and hepatocellular carcinoma human cell lines, caused the down-regulation of miR-7 targets, reinforcing the hypothesis of a sponge activity of CDR1as on miR-7. Interestingly, Piwecka et al. generated a knocked-out (KO) mouse for CDR1as, expressing a phenotype related to neuropsychiatric disorders, with an altered excitatory synaptic transmission and defects in sensomotor gating [29]. Surprisingly, the authors found that in the CDR1as KO mouse there is a decreased expression of miR-7 and an up-regulation of its mRNA targets. Subsequently, Kleaveland et al. discovered a molecular regulatory network that includes CDR1as, miR-7 and the long non-coding (ncRNA) Cyrano, that could explain the results obtained with the CDR1as KO mouse: in the absence of CDR1as, miR-7 is free to bind to Cyrano, which triggers its degradation [30]. This phenomenon will be discussed later in this review. In the same study, Hansen et al. identified another circRNA that acts as a miRNA ceRNA, represented by the testis-specific circRNA Sry with sponge activity for miR-138 [37]. Those findings led to the hypothesis that the miRNA sponge effects achieved by circRNAs are a general phenomenon.

Nevertheless, genome-level analysis reported weak circRNA enrichment for miRNA binding sites [17]. In addition, circRNA-specific exons are associated to AGO2 less frequently than those presented exclusively in linear isoforms [13,23]. Another important issue in contrast to the ceRNA hypothesis, is that often the expression of the circRNA containing miRNA binding site is not so high as to potentially affect miRNA activity. Although there are questions as to the efficacy of some circRNA to regulate the levels of miRNAs, the miRNA sponge activity is the most studied and characterized function [7,8,37,38,39] and in the literature there are several reports proposing that circRNAs can regulate gene expression by binding to miRNAs [40,41,42]. 

Apart from ceRNA activity, circRNAs binding to miRNAs might express other functions, for instance as regulators of miRNA stability, transport, degradation or storage. We will further discuss some of those potential circRNA activities.

### 3.3. circRNAs as Protein Decoys, Scaffolds for Multiprotein Complex Assembly and Molecular Transporters

circRNAs can efficiently bind proteins in their mature sequence as well as in intronic sequences of the pre-RNA transcript [43,44,45,46,47,48,49,50,51,52,53] To date, the best example of biochemical and functional characterization of circRNA–protein interaction is the highly conserved binding of circMbl to the splicing factor MBNL in Drosophila and in human. The binding of MBNL to introns flanking circMbl promotes the backsplicing. The circRNA biogenesis counteracts mRNA maturation, therefore the level of endogenous MBNL protein controls its production. On the other hand, circMbl functions as a decoy for the RBP, promoting the linear splicing of the gene [43]. This example well defines an autoregulatory circuit in which a circRNA post-transcriptionally tunes the expression of its host gene [25]. Similarly, circPABPN1 derived from the poly(A)-binding protein nuclear 1 gene (PABPN1), binds HuR (ELAVL1), an RBP that activates the translation of PABPN1 mRNA, negatively regulating PABPN1 protein synthesis in Hela cells [44]. circANRIL binds pescadillo homologue 1 (PES1), an RBP implicated in pre-rRNA biogenesis and modulating rRNA maturation [46]. An interesting protein decoy mechanism is the one described for the nuclear circular RNA antagonist for cGAS (cia-cGAS), that competes with self-genomic DNA for the binding to cyclic GMP-AMP (cGAMP) synthase (cGAS) in hematopoietic stem cells (HSCs) homeostasis [47]. The novelty of this regulatory circuit will be discussed in the paragraph new functions.

The ability of circRNAs to function as a protein scaffold is exemplified by circFoxo3 and circAmotl1 [49,51]. In human cells, circFoxo3, interacting with both cyclin-dependent kinase inhibitor 1 (p21) and cyclin-dependent kinase 2 (CDK2), acts as a molecular platform for p21 and CDH2 co-localization by the formation of a ternary complex [49]. The binding of circAmotl1 to 3-phosphoinositide-dependent protein kinase 1 (PDK1) and protein kinase B (AKT1) facilitates the PDK1-dependent phosphorylation of AKT1 [51].

As discussed above, circRNAs may act as molecular transporters recruiting macromolecules to specific cellular locations. An interesting example is the above-mentioned nuclear circFECR1, that recruits TET1 to the promoter of its host gene [36].

Recognition sequence for RBPs might be identified bioinformatically, using algorithms initially developed for linear transcripts [54]. However, biochemical validation of RNA-protein complexes is mandatory, since RBP consensus are weak and parameters developed for linear RNA might not be appropriate for circular molecules. The development of more sensitive technologies for circRNA detection and localization would allow the identification and characterization of new circRNAs functioning as carrier and molecular reservoirs of proteins and miRNAs.

### 3.4. Translatable circRNAs

Since circular RNA molecules lack 5′ cap and poly(A) tail, in order to be translated they require alternative signals to be recognized by ribosomal minor subunits. The translation of circRNAs through internal ribosome entry sites (IRESs) was initially demonstrated in 1995 on synthetic circular molecules [55] and recently confirmed on engineered templates by other studies [56,57,58]. Bioinformatic analysis predicted that thousands of circRNAs carry a putative open reading frame (ORF) downstream of IRES signals [59]. However, to date, the examples of circRNAs translated in vivo are still few [58,60,61,62,63,64,65]. The demonstration of circRNA translation is based on ribosome footprinting, mass spectrometry (MS) and cellular assays [56,57,58,60,61,62,63,64,65]. Pamudurti et al. demonstrated that, in fly heads and mammalian muscle cells, a subset of circRNAs are translated in a cap-independent manner and tend to share a start codon with the hosting RNA, carrying a unique stop codon [61]. Accordingly, the predicted peptides are identical to the N-terminal region of host gene protein product. circRNA-derived truncated proteins might have their own function, or they might act as endogenous competitors of the full-length protein [62]. By MS, Yang et al. unambiguously identified in human cells circRNA-encoded products in human cells, as peptides encoded by backsplicing circRNA junctions [63]. Moreover, the authors showed that N6 -methyladenosine (m6 A), the most abundant RNA base modification, promotes the efficient initiation of protein translation from circRNAs. Remarkably, the m6 A-driven translation of circRNAs seems to be a widespread phenomenon in mammals [63,66]. circRNA translation might play an important role in neuronal functions since the identified circRNA-encoded peptide, the circ-Mbl1-peptide, is enriched in synaptosome fractions [61]. However, factors involved in the circRNA translation process need to be further characterized.

## 4. Technical Approaches of circRNA Analysis

The function of most circRNAs remain unknown. However, the growing interest of the scientific community in dissecting their biological activities has led to the development of various and sophisticated methodological approaches. In this review, we summarize the main biochemical approaches applied to date for the characterization and validation of circRNA molecular functions (Table 1). Only the most representative studies are reported.

### 4.1. Approaches for Functional Genetic Analysis

Altering the expression of circRNAs is a powerful strategy to investigate their function. As a matter of fact, loss-of-function studies by circRNA KD or KO, and gain-of-function studies by circRNA OE represent genetic approaches largely adopted to study their physiological activity.

#### 4.1.1. In Vitro Strategies for circRNA KD and KO

RNA silencing (RNAi) using synthetic small interfering RNAs (siRNAs) targeting the backsplicing junction (BSJ) is a common cellular approach to KD circRNAs in cells [41,51,62]. To improve the efficiency of RNAi, chemically modified siRNAs, Locked Nucleic Acids (LNAs) and Unlocked Nucleic Acids (UNAs), can be used [67]. To overcome the problem of the transient circRNA KD achieved by siRNA transfection, vectors expressing short hairpin RNAs (shRNAs) targeting the BSJ have been designed to be transfected in cultured cells, allowing the specific long-term KD of circRNA expression [40,60]. One of the main problems of circRNA KD is the off-target effect, since linear cognate mRNA or other unrelated transcripts might be silenced as well [68]. More details can be found in recent reviews focused on technical issues related to circRNA analysis [69,70,71]. CRISPR-Cas genome-engineering techniques, in particular the CRISPR-Cas13 system, have been successfully utilized to genetically abolish circRNA formation [72]. Li et al. constructed a library consisting of five guide RNAs (gRNAs) of variable length (26–30 nt), targeting the BSJ of each circRNA. By using the RNA-targeting type VI CRISPR RNase RfxCas13d, they developed a powerful and specific cell system aimed at identifying the circRNAs involved in cell proliferation [73]. To our knowledge, this is the first case in which a high-throughput loss-of-function screening of circRNAs, based on RfxCas13d/BSJ-gRNA system, has been performed in human cells.

#### 4.1.2. In Vivo Strategy for circRNA KD and KO

Two recent studies provided new approaches to KD circRNAs in vivo in animal models [74,75]. Pamudurti et al. generated a Drosophila-specific shRNA vector based on the miR-1 precursor, expressing siRNAs directed against the BSJ of selected circRNAs [74]. The shRNA expression is temporally and spatially controlled by means of the GAL4/UAS expression system. Similarly, Zimmermann et al. achieved a region-specific KD of circHomer1a in the mouse brain by using an shRNA expressed under the human synapsin promoter from a lentiviral vector [75]. Both approaches have low off-target effects.

In vivo circRNA KO have been generated in animal models by using CRISPR-Cas9 technology [29,76]. The genetic deletion of CDR1as in mouse was obtained by removing the circRNA-producing exon. In theory, this approach would potentially affect the expression of the host gene CDR1, located on the opposite strand. However, since no detectable expression of CDR1 mRNA was observed in mouse, the CDR1as KO mouse phenotype could be ascribed only to the disruption of the circular gene product [29]. Parker and colleagues, by analyzing transcriptomic data from the CDR1as KO mouse, observed an up-regulation of the upstream and downstream Riken transcripts, from the CDR1as genomic locus [76]. Therefore, more experiments would be required to clarify the role of CDR1as in different tissues and developmental stages. Another CRISPR-Cas9-based strategy was adopted to deplete the mouse circRNA cia-cGAS in vitro [47]. The KO mouse was generated by removing one of the complementary sequences in the cia-cGAS flanking introns, without affecting the expression of the cognate mRNA. In the future, a better knowledge of regulatory signals exclusive to circRNA biogenesis would allow for the design of highly specific and efficient KD/KO constructs.

#### 4.1.3. Overexpression Studies

Considering that the majority of circRNAs are underrepresented in cells [7], their OE could be a useful method to investigate their biological function [25,37,41,77]. Entire host gene or mini-gene-based constructs can be expressed in cells, and upon processing by the cellular splicing machinery they generate high levels of exonic circRNA molecules [77]. The mini-gene comprises circRNA exons flanked by cis regulatory elements that mediate the circularization. Such elements contain inverted complementary sequences and the endogenous intronic sequences that include splicing sites [18,27,78,79,80]. Trans acting factors, such as RBPs, can favor the proximity of circularized exons [26], therefore the inclusion of RBP-binding motifs in flanking introns can promote circRNA biogenesis and expression. It should be pointed out that circRNA OE from mini-gene constructs could produce RNA concatemers and could generate circRNA expression levels many folds above the physiological level. This could be considered an important caveat for circRNA-based functional studies [77].

To study the coding potential of circRNAs, Yang and Wang developed a GFP-based mini-gene reporter, in which the GFP ORF can be reconstituted by the backsplicing reaction, allowing reporter expression [81].

## 5. Neuronal circRNAs

AS strongly contributes to the mammalian transcriptome complexity, especially in the nervous system [82]. More than 90% of human protein-coding genes produce multiple mRNA isoforms; and also non-coding gene transcripts are extensively alternative spliced [82,83,84,85]. Neuronal genes generate a huge variety of transcripts, often functionally distinct [82]. It is well established that neurons make extensive use of AS, and in general of post-transcriptional mechanisms of gene expression regulation, to tightly orchestrate complex biological processes involved in neuronal development and activity. An emerging theme in molecular neurobiology is that RNA processing, and particularly splicing mis-regulation, plays a crucial role in the etiology of neurological diseases [86].

circRNAs can be considered newly characterized splicing isoforms, with functional relevance. As already mentioned, the mammalian as well as the fly brains are unique among all other organs owing to the abundance of circRNAs [6,8,11,12,13,23]. Their extraordinary complexity in the nervous system might be related to the highly regulated apparatus for RNA splicing in the brain. Neuronal circRNAs are conserved in mammals over evolutionary time, both in sequence and expression patterns [12]. They are differentially expressed throughout the brain regions, with an overall enrichment in the cerebellum [12]. Brain-expressed circRNAs are mainly exonic, with a clear preference for coding sequence and 5′ UTR exons [12]. They mainly derive from genes for neuronal proteins that represent almost 20% of the protein-coding genes [13]. Interestingly, most of circRNA host genes show an overall complex pattern of AS. A recent study on a large discovery dataset clearly indicates that the percentage of alternative exons is significantly higher among circRNA-forming exons, than among non-circRNA-forming exons [87]. The relative ratio between circular and linear transcripts is generally higher in the adult brain than in other tissues [12,13]. A striking example is the mouse circRims2, whose expression in the mouse brain is 20-fold higher than the linear mRNA, whereas it is lowly expressed in other tissues [12]. In conclusion, a highly regulated alternative processing of RNA primary transcripts might explain the extraordinary abundance of circRNAs in the nervous system. Moreover, the post-mitotic nature of the adult brain, in which tissue renewal is low, can further contribute to circRNA accumulation.

### 5.1. circRNAs in Brain Development and Homeostasis

Neuronal circRNAs are developmentally regulated [12,13,16,87] and during rodent, pig and fetal human development a global trend of increasing circRNA expression was observed, with a region specificity [12,88]. In the frontal cortex of the human fetal brain, more than in other brain regions, hundreds of genes have dominant circular isoforms [88]. A clear dichotomy in the regulation of the expression of circular and linear transcripts, deriving from the same gene, has been demonstrated by the comparative analysis of circRNA and mRNA expression, from early and late developmental stages of the human brain [14]. Interestingly, gender-associated differences have been observed in the rat developing brain [88,89]. During neuronal differentiation and maturation there is an overall enrichment of circRNAs at the synapse and dendritic arbors [12,13]. Moreover, the vast majority of neuronal circRNAs derive from genes coding for synaptic proteins [13] and their expression is regulated by neuronal activation and homeostatic plasticity [90]. The above features of neuronal circRNAs strongly suggest their involvement in synaptic functions [12].

### 5.2. circRNAs in Ageing

circRNAs accumulate during brain ageing. This phenomenon, originally observed in Drosophila [11], has later been described in other animals as well [15,89,91]. During aging, the observed accumulation of circRNAs in brain tissues was found to be largely independent of the cognate linear transcript [15]. The exceptional stability of circRNAs might explain the age-accumulation trends observed in neural tissues, which have a high composition of post-mitotic cells [91].

### 5.3. circRNAs in Neuronal Disorders

Changes in the level of circRNAs have been observed in human neuronal disorders and human disease animal models (Table 2)

Altered circRNA expression is often region-specific [87,92] and independent of the linear gene products [14,87,92,93]. Since circRNA abundance depends on the circRNA formation, the altered circRNA levels observed might depend on a deregulation of their biogenesis. As discussed above, circRNA formation is regulated by cis- and trans-regulatory factors.

Cis-acting elements are sequence motives and secondary structures in the precursor RNA, implicated in splicing regulation and circRNA formation. Therefore, mutations in cis-acting elements could be involved in disease-associated circRNA deregulation. Interestingly, Liu and co-authors have recently identified circRNA quantitative trait loci (circQTLs) by combining the expression level of human prefrontal cortex circRNAs with genetic SNPs. The authors suggest that partial circQTL SNPs could affect circRNA formation by altering splicing sites, and consequently circRNA biogenesis. A subset of these circQTL SNPs is highly associated with schizophrenia [94]. Trans-factors are general splicing regulators and RBPs, that might also determine the balance between circRNA biogenesis and canonical splicing [25]. Alterations in the level and function of splicing regulatory proteins have been described in a number of neurologic diseases: MBL in myotonic dystrophy (MD), Fused in Sarcoma (FUS) and 43-kDa transactivating responsive sequence DNA-binding protein (TDP-43) in Amyotrophic Lateral Sclerosis (ALS) and QKI in schizophrenia [95,96,97,98,99] have been demonstrated to be disease determining factors. As all of the above-mentioned RBPs are implicated in circRNA biogenesis [25,26,100] general defects in AS might be associated with an overall circRNA deregulation. As a proof of concept, in the frontal cortex of schizophrenic human patients [75,101] and in ALS patients [102] a broad circRNA dysregulation has been observed. In general, disease-associated circRNAs are considered promising biomarkers of the disorder, since they can be measured in different biofluids such as peripheral blood and saliva [8,103]. As discussed above, circular RNAs are enriched and stable in exosomes [104,105], extracellular vesicles which are present in different biofluids, such as saliva, urine and plasma [106]. Exosomes can pass through the blood–brain barrier in the presence of neuroinflammation, often implicated in psychiatric disorders such as schizophrenia, bipolar disorder, and major depressive disorder. Therefore, exosomal circRNAs are considered ideal biomarkers for such diseases [105,107,108].

The major question arising from the finding of circRNA dysregulations in neuronal pathologies is their direct implication in the disease etiology. The answer to this question implies the knowledge of circRNA molecular function. On the other hand, the study of disease-related circRNAs can be helpful for the characterization of their molecular function. It should be pointed out that circRNAs with a well-established molecular function are currently the minority. Interestingly, circHomer1, a highly expressed circRNA in the mammalian brain, was found to be significantly reduced in the prefrontal cortex of psychiatric patients suffering from schizophrenia and bipolar disorder, and in a patient iPS-derived neuronal culture [75]. The in vitro KD of circHomer1 in the mouse prefrontal cortex strongly altered the AS of genes involved in synaptic plasticity and psychiatric diseases. Behavioral cognitive deficits were also observed in the KD mice. The authors demonstrated the binding of circHomer1 to the RBP HUD, known to be implicated in synaptic plasticity, AS and RNA transport. circRNA decoy activity for HUD might suggest a role of circHomer1 in the regulation of alternative splicing in normal and pathological conditions. Another recent study shows that the differential expression of circRNA observed in the frontal gyrus of patients with bipolar disorder is associated with an overall dysregulation of AS [109]. In the overall, studies reported in Table 2 suggest that, in psychiatric disorders, a circRNA general dysregulation correlates with a global change in AS [87,110,111]. Various Aβ circRNAs are generated from the human APP gene and are potentially translated [112,113]. Mo and co-authors showed by in vitro studies that the Aβ circRNAs OE determines the formation of Aβ plaques outside of neuron [112].

Most studies on circRNA expression in neuronal diseases correlates the circRNA expression profile with miRNA and mRNA profiles (Table 2). Disease-regulatory networks of circRNA-miRNA-mRNA are bioinformatically extracted, by the identification of microRNAs targeting the circRNA sequences with an opposite differential expression, and the miRNA target mRNA with a regulation similar to circRNA [81,90,93,114,115,116,117,118,119,120,121,122,123]. Based on the presence of one to seven miRNA target sequences on the circRNA, the decrease in miRNA levels is associated with the observed increase in the cognate circRNA levels through a sponge mechanism, in which circRNAs act as competing endogenous RNAs [124,125,126,127]. In some studies, the predicted circRNA–miRNA interaction is validated by reporter assay. However, as already discussed by others, it is unlikely that a single or even multiple miRNA binding sites on a circRNA molecule are able to alter the levels of functional miRNAs. It should be noted that the miRNA sponging activity of circRNAs, containing a binding sequence for miRNA seed region, would hardly alter free miRNAs, detected by sequencing analysis. The circRNA CDR1as is per se an exception, containing 130 and 73 recognition sites to miR-7, in mouse and human Cdr1as, respectively [8,37]. Interestingly, reduced levels of CDR1as have been reported in patients suffering from Alzheimer’s disease [128]. We will discuss later alternative the biological implications of a subset of circRNA–miRNA interactions.

**Table 2 cells-09-01815-t002:** Outline of circRNAs associated to neurological diseases.

Disease	Model System	circRNA (Validated)	circRNA Profile	Predicted Mechanism of Action	Putative Regulatory Network and Molecular Implication	Refs
SCZ	Human DLPFC	circGPR137B-3, circPPP2CA-3, circVCAN-2, circLONP2–6, circTOP1–10, circMYO9A-66, circHP1BP3–7 and circZNF236–2 down	Overall decrease of circRNA expression and complexity	miRNA sponge	circRNA-miRNA-mRNA network circRNA differential expression independent from host gene expression Decreased AS	[101]
SCZ and AFF	Human DLPFC		Identification of SNPs in circQTLs		Effects on circRNA biogenesis	[94]
SCZ and BPD	Human DLPFC, OFC and iPS-derived neurons	circHomer1 down in SCZ and AD; circCUL4A up in SCZ only; circADAM2 down in BPD only	Overall circRNA dysregulation	RBP decoy: circHomer1/HuD	Altered AS of synaptic genes and psychiatric disease associated genes	[75]
BPD	Human MFG	cNEBL and cEPHA3 up	No overall dysregulation of circRNA expression		Overall increased number of AS events and genes carrying AS	[109]
ASD	Human FC, TC and CV	circARID1A up	60 dysregulated circRNAs	miRNA sponge: circARID1A/miR-204-3p	circRNA-miRNA-mRNA network circRNA differential expression independent from host gene expression	[119]
ASD	Human FC, TC and CB	circZKSCAN1, circHIPK3	ASD specific circRNA co-expression module in FC	miRNA sponge: CDR1as/miR-7	CDR1as-miR-7-miR-671-Cyrano network. Percent of AS exons higher among circRNA-forming exons	[87]
ASD	Mouse HP from the BTBR T + tf/J ASD model	circCdh9, circCacna1c, circCacna1a, circHivep2, circCdc14b, circTrpc6, circCep112, circWdr49, circNcoa2 down; circRmst, circZcchc11, circMyrip up	41 dysregulated circRNAs		circRNA differential expression independent from host gene expression	[92]
AD	Human PC	circHOMER1 and circKCNN2 down	33 dysregulated circRNAs	miRNA sponge	circRNA-miRNA-mRNA network circRNA differential expression independent from host gene expression	[120]
AD	Human HP CA1 and neo C from SAD patients	CDR1as down		miRNA sponge: CDR1as/miR-7	CDR1as-miR-7-UBE2A mRNA network	[117,128]
AD	Mouse brain from SAMP8 model		235 dysregulated circRNAs	miRNA sponge	circRNA-miRNA-mRNA network	[115]
AD	Rat HP from β_1-42_-induced AD model		555 dysregulated circRNAs	miRNA sponge	circRNA-miRNA-mRNA network	[113]
AD	Human HEK293 cells	17 circRNAs derived from Aβ ORF of the APP gene		Peptide synthesis	Increases levels of Aβ and Aβ plaques	[112]
PD	Mouse midbrain from MPTP-PD model and human MPP^+^-SH-SY-5Y cell PD model	circDLGAP4 down		miRNA sponge	circDLGAP4-miR-134-5p-CREB mRNA network	[116]
PD	Human MPP+ SH-SY-5Y cellular PD model	circSNCA up		miRNA sponge	circSNCA-miR-7-SNCA mRNA network	[121]
MMD	Human blood		146 dysregulated circRNAs	miRNA sponge	circRNA-miRNA network	[117]
MS	Human PBL	circ_0005402 and circ_0035560 down	406 dysregulated circRNAs		New biomarkers	[108]
ALS	Human PBMCs	circ_0023919, circ_0063411 and circ_0088036	425 dysregulated circRNAs		New biomarkers	[102]
TL Epilepsy	Human TC	circEFCAB2, circSTK24, circVPS37C up; circDROSHA, circSTK17A, circUBQLN1, circCCT4, circUSP9 down	442 dysregulated circRNAs	miRNA sponge	circRNA-miRNA-mRNA network prediction	[118]
TL Epilepsy	Human TC	circRNA-0067835	586 dysregulated circRNAs	miRNA sponge	circRNA-0067835-miR-155-FOXO3a mRNA network	[122]
Epilepsy	Mouse pilocarpine chronic epilepsy model		43 dysregulated circRNAs	miRNA sponge	circRNA-miRNA-mRNA network	[123]

**Abbreviations:** SCZ: Schizophrenia; AFF: Affective/mood disorder; BPD: Bipolar Disorder; ASD: Autism Spectrum Disorder; AD: Alzheimer Disease; SAD: Sporadic Alzheimer Disease; ADAD: Autosomal Dominant AD; PD: Parkinson’s Disease; MMD: Moyamoya Disease; MS: Multiple Sclerosis; ALS: Amyotrophic lateral sclerosis; TL Epilepsy: Temporal Lobe Epilepsy; DLPFC: Dorsolateral Prefrontal Cortex; OFC: Orbitofrontal Cortex; iPS: Induced Pluripotent Stem; MFG: Medial Frontal Gyrus; FC: Frontal Cortex; TC: Temporal Cortex; CV: Cerebellar Vermis; CB: Cerebellum; HP: Hippocampus; PC: Parietal Cortex; SAMP8: Senescence Accelerated Mouse Prone 8 Model; MPTP (1-metil 4-fenil 1,2,3,6-tetraidro-piridina); MPP+: 1-methyl-4-phenylpyridinium; PBL: Peripheral Blood Leucocytes; PBMC: Peripheral Blood Mononuclear Cell; SNP: Single-Nucleotide Polymorphism; AS: Alternative Splicing; MBS: miRNA Binding Sites; RBP: RNA Binding Protein; circQTLs: circRNA Quantitative Trait Loci; down and up: circRNA down-regulated and up-regulated.

## 6. circRNA New Functions

RNA structure and function studies taught us that high-order RNA structures confer extra layers of functionality to ribonucleotide sequences [129]. Owing to their covalently closed conformation, a circRNA has unique structure, distinct from that of the linear isoform. In fact, a circular RNA molecule can create a constraint on RNA folding which may induce new tertiary structures, generating different motifs beneficial in certain contexts [130]. We can imagine that those conformational patterns are new recognition motifs for biological molecules, or that they could represent higher affinity recognition signals for target macromolecules than the one present in the linear isoform. Such mechanisms would biochemically support the activity of circRNAs as ceRNAs.

As a relevant example of a ceRNA, the cia-cGAS circRNA inhibits the cyclic GMP-AMP (cGAMP) synthase (cGAS) by blocking its enzymatic activity to maintain cellular homeostasis. cGAS is a DNA sensor and its catalysis is induced by binding to DNA, which ultimately elicits the production of type I IFNs. Since cia-cGAS harbors a stronger binding affinity to cGAS than self-DNA does, this circRNA consequently suppressed cGAS-mediated production of type I IFNs in long-term hematopoietic stem cells [47] Intra-double-strand RNA (dsRNA) motives, 16–26 base pair duplexes, present in cellular circRNAs represent previously unknown inhibitors for PKR in cells related to innate immunity [32]. Remarkably, high-order structures specific of circRNAs are involved in circRNA decay [131]. Fischer et al. recently reported a circRNA decay mechanism that selectively degrades highly structured RNAs, sensing overall RNA structure rather than defined sequences. The so-called structure-mediated RNA decay (SRD) requires the RNA-binding protein UPF1 and its associated protein G3BP1, and was predicted to regulate up to one-third of circular RNAs in humans [131].

Neuronal circRNAs have been recently proposed as memory molecules. Chen and Schuman recently suggested that, in the brain, circRNAs might function as molecules for long-term memory storage [132]. This fascinating hypothesis relies on the ‘longevity’ of circRNAs and on their presence in specific neuronal compartments, such as dendrites [12,13]. From a molecular point of view, by binding small RNAs and proteins related to neuronal plasticity, they could act as a source of memory retention at the synapses.

## 7. Perspectives and Conclusions

miRNAs with extensive complementarity to target RNAs are degraded through a mechanism called target-directed miRNA degradation (TDMD). In TDMD, the extensive complementarity of the 3′ region of the miRNA to the target RNA expose the miRNA 3′ end to tailing (3′ addition of non-templated nucleotides, Us or As), trimming (shortening from the 3′ end) and degradation [133]. This mechanism was initially described to be induced by viral RNAs targeting host miRNAs to degradation [134,135]. Subsequent studies indicated that TDMD is not specific to viral RNAs but is also sustained by eukaryotic RNAs and conserved from fly to human [136,137]. In rodent neurons, TDMD is particularly effective, highly contributing to the dynamic regulation of neuronal miRNAs [138]. De la Mata et al. accurately and quantitatively characterized the TDMD mechanism in neurons, by designing synthetic targets to the neuronal miR-132 and miR-124. The authors showed that, unlike mRNA silencing, neuronal TDMD works as a multiple turnover and does not involve co-degradation of the target. Moreover, TDMD does not rely on cooperativity among multiple target sites to reach high efficacy [138]. Remarkably, an endogenous TDMD event and its implication for normal animal behavior was recently described in zebrafish and mouse, in which the transcript of the NREP gene triggers miR-29 degradation [139]. An extensive complementarity between the mouse long ncRNA Cyrano and miR-7 triggers the degradation of miR-7 through a TDMD mechanism [30]. By reducing miR-7 levels, Cyrano enables the accumulation of Cdr1as. In the absence of Cyrano, high levels of miR-7 cause the destruction of CDR1as, in part through enhanced slicing of CDR1as by a second miRNA, miR-671 with an almost perfect complementary to CDR1as. Cyrano, together with miR-7 and CDR1as, participates in a highly efficient neuronal regulatory network.

### Can circRNAs Trigger TDMD?

We reasoned that a circular RNA might trigger TDMD as well as linear RNA. Since the high complementarity of the binding of the miRNA to its target is a primary requirement for TDMD to be activated, we performed an in silico analysis searching for interactions between known circRNAs and miRNAs with extensive complementarities at the 5′ and 3′ miRNA ends. Bioinformatic predictions have been obtained by using RNA22 v2 tool (used with default parameters except “Minimum number of paired-up bases in heteroduplex” that was set to 15) [140], structures designed by using RNApdbee 2.0 tool [141]. In addition, as the pathway of TDMD in mammals is particularly active in neurons, we performed our analysis on a list of synaptoneurosomal circRNAs downloaded from circBase (http://www.circbase.org), [12,142] and a list of somato-dendritic miRNAs [143,144,145] (Table 3 and Appendix A). It is worth mentioning that somato-dendritic miRNAs are highly regulated in neuronal tissues and cells [143,146,147].

Interestingly, we found various low free-energy interactions between miRNAs and circRNAs (see Appendix A for the complete analysis). In Table 3 potential circRNA/miRNA heteroduplexes are presented, with binding free energy and secondary structure predictable for TDMD-inducing complexes, such as the one described for Cyrano and miR-7 (shown in Table 3), and other RNAs [30,133,148].

Notably, there are four miRNA/circRNA interactions with a low ΔG (<−25 Kcal/mol), close to the mir-7/Cyrano ΔG value (−28.9 Kcal/mol), and other two (miR-129-5p/mmu_circ_0004245—Sipa1l1 and miR-146a-5p/mmu_circ_0007086—Fbxo11) with a higher ΔG but secondary structures compatible with the proposed model of interaction of miRNA and TDMD, inducing RNA targets [133,148].

According to the TDMD features, we hypothesize that even one single, highly complementary binding site on the circRNA molecule would trigger an efficient degradation of the complementary miRNA. In our model, the efficacy of the circRNA-mediated TDMD process could rely on a multiple turnover activity as well as on the high stability of the target RNA. However, circRNA conformational constraints should be considered, binding sites might not be accessible to miRNAs, and biochemical validation would be mandatory to test such hypothetical interactions and biological outcomes. Figure 2 describes known and putative circRNA molecular functions.

In conclusion, circRNAs, hidden in their molecular complexity for many years and recently described as new regulatory RNAs, are still a secret garden to be discovered [149]. Definitely, in the nervous system, where a tight control of gene expression is required both spatially and temporally, circRNAs represent key regulatory molecules with anticipated molecular functions.

## Figures and Tables

**Figure 1 cells-09-01815-f001:**
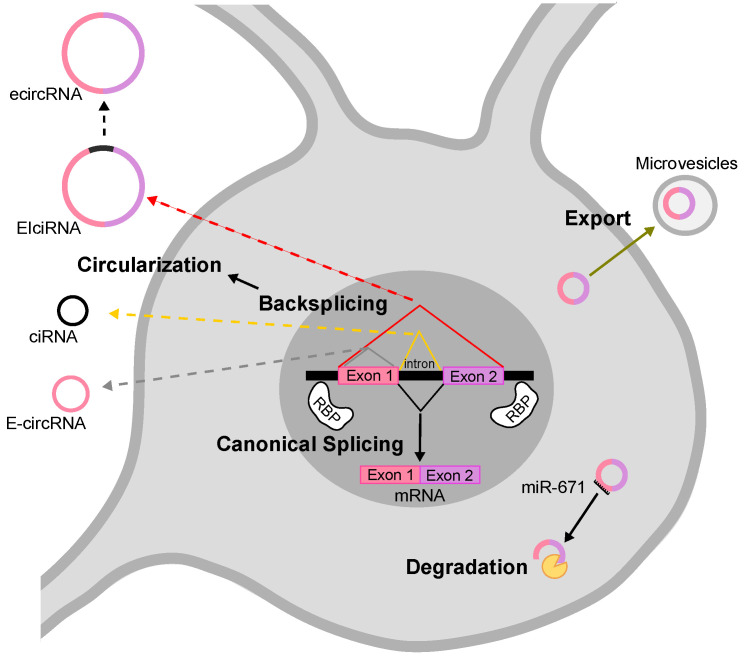
Biogenesis and removal of circRNAs. On the left, circular RNA molecules originating from backsplicing events: E-circRNA from exon1- gray arrow; ciRNA from intron- yellow arrow; EIciRNA from pre-mRNA- red arrow. On the right, known pathways of circRNA elimination from cells: degradation through an almost perfect complementarity with a miRNA (i.e., miR-671) and export through microvesicles and exosomes. E-circRNA: exonic circRNA; ciRNA: intronic circRNA; EIciRNA: exonic-intronic circRNA; ecircRNA: exonic circRNA originating by splicing of EIciRNA; RBP: RNA-Binding Protein.

**Figure 2 cells-09-01815-f002:**
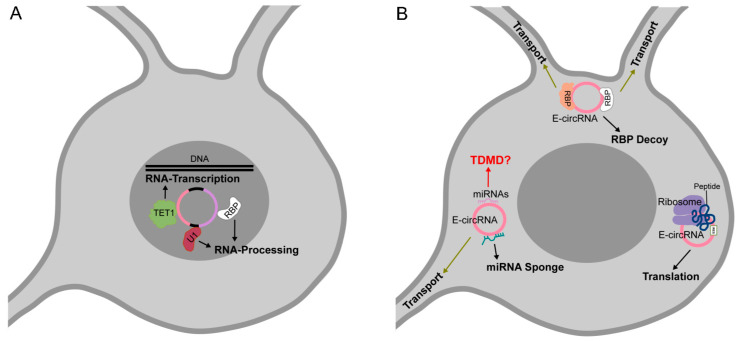
Known and potential new circRNA functions. (**A**) Nuclear circRNA functions: interacting with U1 snRNP and transcription complexes, exon-intron circRNAs (EIciRNAs) can regulate host gene expression and by recruiting TET1 demethylase enzyme can modulate the chromatin epigenetic status; (**B**) Cytoplasmic circRNA functions: circRNAs containing ORF and IRES can be translated. circRNAs containing binding sites for RBPs can function as RBP decoys, competing with RBP targets. By binding to miRNAs or RBPs circRNAs can function as molecular transporters, to localize macromolecules to specific cellular sites and to transport them to neuronal processes. circRNAs containing several binding sites for a particular miRNA can function as a miRNA sponge. We propose that circRNAs, containing a high complementary binding site for a miRNA, might activate TDMD (red question mark). ORF: open reading frame; IRES: internal ribosomal entry site; TDMD: target dependent miRNA degradation.

**Table 1 cells-09-01815-t001:** Methodological approaches for functional characterization of circRNAs.

circRNA	Molecular Mechanism	Validation Method	Putative Physiological Function	Refs
ci-ankrd52	Interacts with the RNA Pol II elongation complex	KD studies RNA/DNA double FISH RIP assay RNA pull-down assay	Promotes transcription of host gene	[18]
circEIF3J	Interacts with U1 snRNP and RNA Pol II	Nuclear Run-On assay ChIRP assay RIP assay RNA pull-down assay	Promotes transcription of host gene	[21]
circSEPALLATA3	Forms R-loops	R-loop dot-blotting	Regulates splicing of cognate mRNA	[35]
FECR1	Interacts with TET1 Binds to the DNMT1 promoter	RAT assay RNA ChIP assay OE studies	Promotes transcription of host gene by inducing DNA hypomethylation	[36]
CDR1as/ciRS-7	miR-7 sponging	Ago PAR-CLIP Ago HITS-CLIP	Neuronal development	[8,37]
circHIPK3	miR-24 sponging	Biotin-coupled miRNA pull-down assay RIP assay Luciferase reporter assay	Regulates cell growth	[41]
circBIRC6	miR-34a and miR-145 sponging	Biotin-coupled miRNA pull-down assay RIP assay Luciferase reporter assay	Promotes pluripotency of human embryonic stem cells	[40]
circRNA.2837	mir-34a sponging	OE and KD studies RIP assay Luciferase reporter assay	Regulates neuronal autophagy	[42]
circANRIL	PES1 decoy activity	λN-Peptide-mediated pull-down assay RIP assay	Impairs pre-rRNA processing and ribosome biogenesis, conferring atheroprotection	[47]
circPABPN1	HuR decoy activity	RIP assay RNA pull-down assay	Suppresses PABPN1 mRNA translation reducing cellular proliferation	[44]
circ-DNMT1	AUF1 and p53 nuclear translocation	Oligo blocking assay RIP assay RNA pull-down assay	Enhances in breast cancer cell proliferation, survival, autophagy and inhibits cell senescence	[51]
circAGO2	HuR decoy activity and cytoplasmic translocation	EMSA RNA pull-down assay Luciferase assay	Enhances cancer progression	[46]
circ-Foxo3	Promotes MDM2 and p53 interaction	KD studies RIP assay RNA pull-down assay	Enhances cell apoptosis	[49]
circ-Amotl1	Promotes AKT1 and PDK1 interaction and nuclear translocation	Oligo blocking assay KD studies RIP assay RNA pull-down assay	Reduces cell apoptosis and promotes cardiac repair	[52]
cia-cGAS	Inhibits cGAS interaction with genomic DNA	RIP assay RNA pull-down assay FISH & IF assay	Contributes to the maintenance of long-term hematopoietic stem cells dormancy	[48]
circYAP	Silences Yap mRNA translation by inhibiting IF4G and PABP interaction	Cap-binding pull down assay IP-IB assays RIP assay RNA pull-down assay	Inhibits cancer progression	[53]
circNOL10	Interacts with and inhibits SCML1 ubiquitination	RNA pull-down assay EMSA Luciferase assay	Reduces cell cycle progression and cell proliferation, and increases apoptosis	[45]
circMbl	Encodes circMbl1-peptide	Ribosome footprinting TRAP assay WB analysis	Putative synaptic functions	[61]
circ-ZNF609	Encodes circ-ZNF609-derived protein	Sucrose gradient fractionation Flagged protein IP WB analysis	Regulates proliferation of myoblasts	[62]
circ-FBXW7	Encodes the FBXW7-185aa protein	Luciferase assay Flagged protein WB analysis LC-MS analysis	Reduces proliferation and cell cycle progression of glioma cell lines	[60]
circPINTexon2	Encodes the PINT87aa protein	RNC-seq IRES activity test Luciferase assay WB analysis LC-MS analysis	Reduces glioblastoma cell proliferation	[64]
circ-AKT3	Encodes the AKT3-174aa protein	Luciferase assay MS followed by SDS-PAGE IF staining analysis	Reduces glioblastoma tumorigenicity	[58]

**Abbreviations:** KD: Knockdown; FISH: Fluorescence In Situ Hybridization; RIP: RNA ImmunoPrecipitation; snRNP: small nuclear Ribosomal Nucleoprotein; ChIRP: Chromatin Isolation by RNA Purification; mRNA: messenger RNA; RAT: Reverse transcription-associated Trap; ChIP: Chromatin Immunoprecipitation; OE: Overexpression; PAR-CLIP: Photo-Activatable Ribonucleoside Cross-Linking and Immunoprecipitation; HITS-CLIP: HighThroughput Sequencing of RNA Isolated by Cross-Linking Immunoprecipitation; rRNA: ribosomal RNA; EMSA: Electrophoretic Mobility Shift Assay; IP-IB: Immunoprecipitation followed by ImmunoBlotting; TRAP: Translating Ribosome Affinity Purification; WB: Western Blotting; IP: ImmunoPrecipitation; LC-MS: Liquid Chromatography–Mass Spectrometry; IRES: Internal Ribosome Entry Site; RNC-seq: Ribosome nascent-chain complex-bound RNA sequencing; MS: Mass spectrometry; SDS-PAGE: Sodium Dodecyl Sulphate—PolyAcrylamide Electrophoresis; IF: ImmunoFluorescence.

**Table 3 cells-09-01815-t003:** Predicted high-energy interactions between somato-dendritic miRNAs and synaptoneurosomal circRNAs.

miRNA/circRNA	ΔG	Secondary Structure
miR-7/Cyrano ncRNA	−28.9	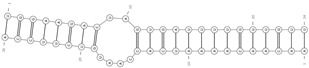
mmu-miR-185-5p/mmu_circ_0001796—Zfp609	−28.2	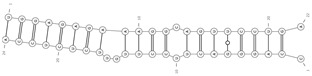
mmu-miR-134-5p/mmu_circ_0005351—Pinx1	−28.0	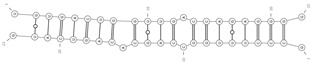
mmu-miR-188-5p/mmu_circ_0000989—Spopl	−26.4	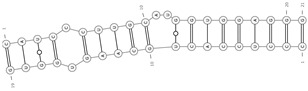
mmu-miR-129-5p/mmu_circ_0005285—Myh6	−25.5	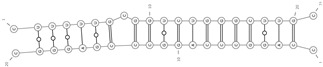
mmu-miR-129-5p/mmu_circ_0004245—Sipa1l1	−24.0	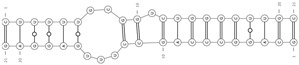
mmu-miR-146a-5p/mmu_circ_0007086—Fbxo11	−22.5	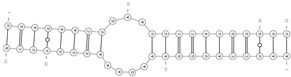

Representative predicted high-energy interactions between somato-dendritic miRNAs and synaptoneurosomal circRNAs are reported. Folding energies (ΔG, Kcal/mol) and heteroduplex secondary structures are indicated. miR-7/Cyrano heteroduplex is also reported. Above is the miRNA sequence 5′–3′, beneath circRNA sequence 3′–5′.

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
