# Peer review of "The Secret Garden of Neuronal circRNAs"

_cells, 2020, doi:10.3390/cells9081815_

Round 1

Reviewer 1 Report

The paper reviews an important area of cellular RNA molecules that potentially play critical roles in the regulation of gene expression in normal and diseased states. It is informative and well organized, however, there are numerous grammatical corrections required. The following are recommended editorial changes, in bold, on noted lines:

22- shortly

25- incorrect splice events was

48- into in

49- locally spatially, in nuclear

64- less well characterized

69- sequences are able

71- Alu repeats, a sequences are very

72- genomes, and are a good examples

76- structures does not always necessarily indicate

83- then thus facilitating

89- RBPs are able...pre-mRNA, that interfere

90/91- (ADAR1), determines results in

95- of many circRNAs

96-48 hours, and up

97- stability, how can cells can eliminate regulate the concentration of their circRNAs?

100/101-mechanisms, endonucleolytic cleavage,

102/103- It is worth mentioning Very recent studies showed

105- for PKR (Protein Kinase-R) activation

106- hand, circRNA the intracellular

169- Although these critical aspects Although there are questions as to the efficacy of some circRNAs to regulate the levels of micRNAs,

245- main problems

259- by means of

268- from the CDR1as

280- high levels of

286- expression levels many folds above the physiological

293-94- not clear what is being stated-poor English

305- are evolutionary conserved over evolutionary time

313-generally higher in

320- are developmentally

321- trend of increasing of circRNA

326- human brain development

371- biofluids such as periferal

375- disorders such as

384-5- in a patient

402- associated with to the

407- It should be noted pointed......containing a binding

420- macromolecules, than

441-444- material OK with reference to the review, however, lines 445-465 deals with information not really relevant to the paper and should be deleted.

I am attaching a copy of the manuscript with all of the "lines" that should be corrected highlighted.

Author Response

Response to Reviewer 1 Comments

Point 1: The paper reviews an important area of cellular RNA molecules that potentially play critical roles in the regulation of gene expression in normal and diseased states. It is informative and well organized, however, there are numerous grammatical corrections required. The following are recommended editorial changes, in bold, on noted lines.

Response 1: According to the Reviewer suggestions, we made all the editorial changes recommended.

Point 2: 441-444- material OK with reference to the review, however, lines 445-465 deals with information not really relevant to the paper and should be deleted.

Response 2: We agree with Reviewer 1, the paragraph (lines 441-465) is too long and some information is not relevant to idescribe the mechanism of TDMD. We removed two sentences "The viral ncRNA HSUR1 from the Herpesvirus saimiri directs a dramatic decrease of the host miR-27 in virally transformed cells. Similarly, the cytomegalovirus MCMV m169 transcript targets miR-27 to degradation via a single binding site in its predicted 3'UTR (free energy of -25.7kcal/mol). In both studies, the viral transcripts direct degradation of mature miR-27 in a sequence-specific and binding-dependent manner". However, we beleive that describing main features of TDMD is necessary to frame a TDMD mechanism in the context of miRNA/circRNA interaction (paragraph 7.1)

Reviewer 2 Report

The manuscript provides an exhaustive and comprehensive overview of the issue. Provides a detailed overview of circRNAs biogenesis and function. Technical approach of circRNA analysis has been discussed in the text and additionally collected in a clear table. Neuronal circRNAs have been described in a separate chapter, and the state of knowledge showing changes of their level in various diseases of the nervous system was presented in an extensive and readable table. Finally, the in silico analysis was performed, where the circRNA interactions with miRNAs were tested in the context of the TDMD mechanism.

I only have a few small comments to the review:

  • Ingeniously abbreviation have been placed under the tables, but not all of them have been explained (for example: CB or BTBR T +tf/J in table 2);
  • The letters in the figures are too small, which makes them difficult to read. Please consider increasing the size of the letters;
  • The analysis of circRNAs interactions with miRNAs was shown by delta G. Shouldn't the p-value (shown in the supplementary materials) also be taken into account?
  • Please check that all references were cited in the text. I couldn't find a few.
  • Title or headline of the table were located on a different side than the rest of the table. It would be more convenient for the readers if the table headline to be on the same page as the rest of the table - especially table 3.

Author Response

Response to Reviewer 2 Comments

Point 1: Ingeniously abbreviation have been placed under the tables, but not all of them have been explained (for example: CB or BTBR T +tf/J in table 2)

Response 1: We carefully checked abbreviations and we added all missing explanations in Table 1 and Table 2

Point 2: The letters in the figures are too small, which makes them difficult to read. Please consider increasing the size of the letters;

Response 2: We increased the size of letters in Fig.1 and Fig.2

Point 3: The analysis of circRNAs interactions with miRNAs was shown by delta G. Shouldn't the p-value (shown in the supplementary materials) also be taken into account?

Response 3: We agree with Reviewer 2, p-values should be mentioned for the in silico analysis as the one reported in Table 3. However, we believed that p-values obtained by using the RNA22 v2 algorithm might not give a reliable statistical significance of TDMD related interactions. Since RNA22 was developed for canonical miRNA/RNA interactions, the given p values are related to the binding of miRNA seed sequences to the miRNA response elements  (MREs) on target RNA. As specified on the RNA22 web site (https://cm.jefferson.edu/rna22-full-sets-of-predictions/),  " ...a lower p-value represents a greater chance that the loci contains a valid MRE". This is the reason why we decided not to report p-values on the table, but only on the Supplementary Table.

Point 4: Please check that all references were cited in the text. I couldn't find a few.

Response 4: Some reference citations were missing, we checked and fixed all of them.

Point 5: Title or headline of the table were located on a different side than the rest of the table. It would be more convenient for the readers if the table headline to be on the same page as the rest of the table - especially table 3.

Response 5: We edited the manuscript to have the table headline on the same page as the rest of the table.